# What are memories made of? A survey of neuroscientists on the structural basis of long-term memory

Ariel Zeleznikow-Johnston[1]*, Emil F. Kendziorra[2], Andrew T. McKenzie[3]*

1 School of Psychological Sciences, Monash University, Melbourne, Australia, 2 European Biostasis Foundation, Riehen, Canton of Basel-Stadt, Switzerland, 3 Apex Neuroscience, Salem, Oregon, United States of America

* arielzj.phd@gmail.com (AZJ); amckenzie@apexneuro.org (ATM)

## Abstract

Despite the last decade's development of optogenetic methods for artificially manipulating engrams, and subsequent claims that there is a consensus that memories are stored in ensembles of synaptic connections, it remains unclear to what degree there truly is unanimity within the neuroscientific community about the neurophysiological basis of long-term memory. We surveyed 312 neuroscientists, comprising one cohort of experts on engram research and another of general neuroscientists, to assess this community's views on how memories are stored. While 70.5% of participants agreed that long-term memories are primarily maintained by neuronal connectivity patterns and synaptic strengths, there was no clear consensus on which specific neurophysiological features or scales are critical for memory storage. Despite this, the median probability estimate that any long-term memories could potentially be extracted from a static snapshot of brain structure was around 40%, which was also the estimate for whether a successful whole brain emulation could theoretically be created from the structure of a preserved brain. When predicting the future feasibility of whole brain emulation, the median participant estimated this would be achieved for *C. elegans* around 2045, mice around 2065, and humans around 2125. Notably, neither research background nor expertise level significantly influenced views on whether memories could be extracted from brain structure alone. Our findings suggest that while most neuroscientists believe memories are stored in structural features of the brain, fundamental questions about the precise physical basis of memory storage remain unresolved. These findings have important implications for both theoretical neuroscience and the development of technologies aimed at preserving or extracting memory-related information.

**Data availability statement:** A list of the survey questions is available here: https://osf.io/agkrn The full set of participant response data is available here: https://osf.io/bas2u

**Funding:** Research funding for this study was supported by a CryoDAO grant (2024.1) The funders had no role in study design, data collection and analysis, decision to publish, or preparation of the manuscript.

**Abbreviations:** ASC, aldehyde-stabilized cryopreservation; COSYNE, computational and systems neuroscience; LTM, long term memory.

## Introduction

Through what physical means do brains retain information for use weeks, months, or years later? Recall of long-term memories – be it recoiling from a distinct taste after a food-induced illness six months prior, or singing the lyrics of a song one hasn't heard in years – enable an animal's behaviors to be shaped by a lifetime's worth of accumulated experience. For this to be possible, earlier experiences must create some kind of 'memory trace' in an animal's brain, meaning there must be some enduring physical record that bridges the gap between an initial experience and its subsequent recall [1]. But among the many potential mechanisms for information storage in the brain, which are actually responsible for long-term memory retention?

There is evidence that robust, long-lived memories are likely retained by relatively stable and static aspects of neurophysiology. Long-term memories can be recalled even after periods of prolonged global neuronal depolarization and inactivity, for example as can occur in deep hypothermic circulatory arrest [2,3]. This suggests that ongoing global electrophysiological activity is not required for the retention of already formed long-term memories. Additionally, while a temporary period of protein synthesis inhibition disrupts memory formation when administered during or shortly after training, it has been found to have no effect on memory recall when administered six or more days after training [4]. This temporal distinction in the mechanisms between memory formation and recall suggests that long-term memory storage depends on stable structural changes rather than ongoing protein synthesis or dynamic neural activity, unlike working and short-term memory.

Over the past century, neuroscientists have suggested a variety of structural candidates for the physical basis of long-term memory. By 'structural,' we refer to relatively stable physical changes to neural components that persist independently of ongoing neural activity or metabolic processes, though these changes may initially require activity-dependent processes to be established. These structural changes exist on a spectrum from modifications of individual proteins and channel properties through to larger-scale alterations in neural connectivity.

A non-exhaustive list of structural candidates includes: synaptic strength alterations mediated by receptor insertion, removal, or subunit modification [5]; synaptogenesis [6]; intracellular phosphorylation, epigenetic, or other molecular modifications [7]; alterations to neuronal excitability through widespread modifications of voltage- or calcium-dependent ion channels, i.e., intrinsic plasticity [8,9]; changes in axonal myelination [10]; modifications of perineuronal nets and the extracellular matrix [11]; and many others. These candidates are not necessarily mutually exclusive, as long-term memory may well depend on a variety of these mechanisms and their interactions. Additionally, these mechanisms may be redundant, with multiple parallel processes capable of maintaining the same memory information even if some mechanisms become temporarily disrupted or degraded.

Breakthroughs in the past decade have seen the search for the neurophysiological substrates responsible for long-term memories, termed *'engrams'*, pursued with increased vigor. These advances were kicked off by the seminal 2012 publication of a method for artificially forcing the recall of a specific memory through tagging and

optogenetically stimulating a subset of hippocampal neurons [12]. Subsequent experiments have since demonstrated selective memory erasure [13], artificial memory formation [14], and artificial memory linkage [15], among other impressive feats. Given that ensembles of hippocampal and cortical neurons have arguably shown to be both necessary (erasure experiments) and sufficient (forced recall and insertion experiments) for memory-related behaviors, it could be argued that the physical basis of long-term memories has now been established. Indeed, in the words of many of the scientists who have furthered engram research in the past decade, "*There is a clear consensus on where the memory engram is stored—specific assemblies of synapses activated or formed during memory acquisition*" [16].

But is it really true that there is now a consensus that long-term memories are stored in synaptic ensembles? For example, consider how dendritic spines on mature hippocampal neurons disappear when exposed to cold temperatures, yet reappear in the same locations upon rewarming [17]. This finding shows that temporary disruptions to fine neural architecture can occur during certain perturbations (e.g., hypothermia) without necessarily destroying long-term memories (as demonstrated in humans who have undergone deep hypothermic circulatory arrest). For some, this may cast doubt upon any claim that dendritic spine positions are the exclusive substrate for long-term memory maintenance.

Regardless, perhaps there is a common understanding among neuroscientists as to which aspects of neurophysiology are critical for long-term memory storage, and which are irrelevant. Some properties—perhaps the specific polymer subunit count of synaptic microtubules, or the exact positions of cholesterol molecules in the neuronal membrane—would presumably be deemed below the physical scale relevant for encoding memories. At the other extreme, some macroscopic neurophysiological properties, like total hippocampal volume, would likely be considered mere epiphenomena of the mechanisms actually responsible for memory storage. If a consensus truly exists, then neuroscientists must have some agreement that somewhere between these lower and upper bounds there lies a 'critical scale', wherein there exists the properties or mechanisms responsible for long-term memory.

To determine whether or not neuroscientists agree on where this critical scale lies, we performed a survey of both neuroscientists generally and neurophysiology-of-memory experts specifically. Expert surveys in other disciplines, such as philosophy [18,19] and artificial intelligence [20], have proven instrumental in ascertaining what is and is not consensus within an academic community. While there exists a survey of the general US population on their beliefs about how memory works [21], and there have been surveys of memory scientists on the psychological properties of memories [22], we are unaware of any previous surveys of the opinions of experts as to the mechanics of long-term memory storage.

This survey was divided into three major components. The first queried participants about whether they believed long-term memories were stored in structural aspects of neurophysiology, and if so, through which physical mechanisms and at which critical scale(s). The second explored what the implications would be of memories being stored structurally, by asking them for their probability estimates as to whether information contained in long-term memories could theoretically be extracted from static brains with ideal morphological preservation. The final section examined a theoretical test for whether long-term memories are stored structurally, by probing participants on whether readout and reinstantiation of these memory-storing neurophysiological structures in another physical form, such as through whole brain emulation [23], could someday provide proof that long-term memories are indeed stored structurally and at a particular critical scale(s).

For questions about the implications of static brain preservation for memory storage, we used aldehyde-stabilized cryopreservation (ASC) of a laboratory animal as a practical example of a preservation method that is thought to maintain ultrastructure with minimal distortions across the entire brain [24]. Additionally, we asked participants to imagine it was performed under ideal conditions and was technically successful, deliberately discarding the fact that procedural variation or errors in the real world may prevent this ideal from being routinely realised in practice [25]. Rather than focusing on these technical preservation challenges, which we acknowledge are immense, we deliberately asked participants to consider memory extraction under optimal preservation conditions to assess their beliefs about the structural basis of memory storage itself. With this approach, our aim was to specifically target participants' views on whether static brain structures

– i.e., non-dynamic physical aspects of the brain that persist independent of ongoing neural activity – may on their own contain sufficient information for memory retrieval, which is the central theoretical question underlying our study.

## Methods

The survey was distributed from August-October 2024 to two separate cohorts of neuroscientists: (1) those who had published research papers directly related to the neurophysiology of memory (Engram Experts); and (2) any attendees with an abstract listed in the Computational and Systems Neuroscience (COSYNE) conference booklets from 2022−2024 (COSYNE Neuroscientists). COSYNE attendees are self-described as those interested in "the exchange of empirical and theoretical approaches to problems in systems neuroscience" [26]. The questions were focused on their beliefs about the physical basis of memory, as well as the implications of these beliefs in various theoretically plausible scenarios. The survey and its implementation were reviewed by the Pearl Institutional Review Board and received an exemption determination (#2024-0303).

### Survey questions

The survey consisted of 28 questions divided into six sections: 'Demographics', 'Structural basis of long-term memories', 'Theoretical implications of memory storage', 'Brain preservation', 'Whole brain emulation feasibility', and 'Familiarity & comfort with the topics discussed'. Most questions were mandatory for completion, except those that asked participants to optionally provide additional commentary on their responses.

Each of the main sections (i.e., all excluding 'Demographics' and 'Familiarity') were preceded by a page of information providing contextual information and definitions required for the questions that followed. The full text of the survey can be found here: https://osf.io/agkrn

### Recruitment

We recruited the Engram Experts and COSYNE Neuroscientists cohorts separately.

Email addresses for the Engram Experts cohort were obtained in three ways: corresponding authors whose email addresses were listed in a Pubmed search for 'engram' conducted on 2024-06-20; those in a Pubmed search for '(volume electron microscopy) AND (learning OR memory)' conducted on 2024-08-01; as well as all author emails listed in the book *Engrams* [27]. The Pubmed searches used an earliest cutoff date of 2012, and only results for papers written in English were examined. We manually removed papers that were clearly not relevant to neurophysiology (e.g., articles on 'immunological engrams'). After removing duplicates, this yielded 305 unique names and corresponding email addresses.

Email addresses for the COSYNE Neuroscientists cohort were extracted from the COSYNE 2024, 2023, and 2022 conference booklets. Specifically, every attendee whose email address was listed in the abstracts sections of the booklets was included. Potential participants who were already included in the Engrams Experts cohort were removed from this group. After additionally removing duplicates, this yielded 4125 unique names and corresponding email addresses.

### Fielding

Participants were invited to participate in an online survey conducted via the Survey Monkey platform, delivered via private link in an email. The invitation email informed participants the survey would take approximately 20 minutes, and offered a reward on completion of $75 (Engram Experts) or $20 (COSYNE Neuroscientists). Payments were provided via electronic gift-cards by means of the third-party provider Tremendous.

The Engram Experts cohort was sent an initial invitation on August 29, with subsequent reminders on September 4, 9, 14, and 19. The survey remained open until September 20. Out of the 305 emails we contacted, 32 (10.5%) bounced or failed, leaving 273 functional addresses. We received 33 responses, for a response rate of 12.1%. 25 (76%) of

respondents completed the entire survey, defined as providing a response to every mandatory question. To investigate the effect of incentives on response rate, 10% of the collected emails from the COSYNE Neuroscientists cohort were initially assigned to a pilot group that were not offered a reward. The pilot invitation was on September 30. Because this unpaid pilot group had substantially lower response rates than the paid groups, we decided to roll this pilot group into the main cohort and offer a payment to all participants (including those who had initially been in the unpaid pilot group).

The COSYNE Neuroscientists cohort was sent an initial invitation on October 3, with subsequent reminders on October 8 and 14. The survey remained open until October 16. Out of the 4125 emails we contacted, 295 (7.2%) bounced or failed, leaving 3830 functional addresses. We received 279 responses, for a response rate of 7.3%. 205 (73%) of these respondents completed the entire survey.

Participants who completed only the demographic section of the survey did not have a statistically significant difference in their demographic profile from those who completed the entire survey (S1 Fig). Specifically, completers vs non-completers did not differ significantly in terms of age (X: 10.384, df = 7, p = 0.17); b) publication count (X: 10.204, df = 5, p = 0.07); c) education level (X: 4.309, df = 3, p = 0.23); d) or research approach (X: 4.518, df = 2, p = 0.10).

## Analysis

Data analysis was performed using R statistical software, version 4.3.1. Statistical analyses included Hartigans' dip test for multimodality, Spearman correlation analyses, Wilcoxon rank-sum tests for pairwise comparisons, and Kruskal-Wallis tests for group comparisons. All reported p-values are two-tailed.

To assess the relationships between participants' responses to different questions, we performed a rank correlation analysis across the relevant variables in our dataset that we could include. These included demographic variables (age, education level, PhD status, research approach, publication count), self-reported expertise ratings (memory expertise, preservation expertise, neural modeling expertise; each measured on a continuous scale from 1–5), and survey responses (theoretical possibility ratings, ASC probability estimates, emulation probability estimates, timeline predictions, etc.). In order to include them in this correlation analysis, ordinal variables were coded numerically. For example, research approach was coded as +1 (wet-lab), 0 (mixed), and −1 (dry-lab), while memory expertise was coded from 1–4 (not familiar to highly familiar). The cohort classification (engram expert vs. COSYNE neuroscientist) was analyzed separately in the subgroup analysis, not in this correlation matrix. After calculating correlation coefficients and corresponding p-values for all variable pairs, we applied a Benjamini-Hochberg correction to control for multiple comparisons with a false discovery rate of 0.05. Only the correlations that remained significant after this correction were shown in the figure.

## Results

### Beliefs on the physical basis of memory

Participants were initially asked whether they believed it would be theoretically possible, in principle, to extract the information corresponding to a non-trivial, specific, long-term memory from a static map of synaptic connectivity. This question followed after context that defined a 'static map of synaptic connectivity' to include everything down to the level of the states of individual biomolecules, and a 'non-trivial memory' by reference to examples like 'the route for navigating a maze' or 'the memory of a password'. 45.6% of participants agreed or strongly agreed that this would be possible in-principle, given sufficiently sophisticated scientific techniques, while 32.1% disagreed or strongly-disagreed (Fig 1a).

When given multiple, non-exclusive choices for what additional information beyond a static snapshot might be required for memory readout, respondents most commonly selected measurements of dynamically changing neuronal activity patterns (46.5%), followed by contextual information about the animal's experiences and mental states (43.0%), and information about sensory input and motor output (37.0%). Less commonly chosen were the need for measurements of chemical

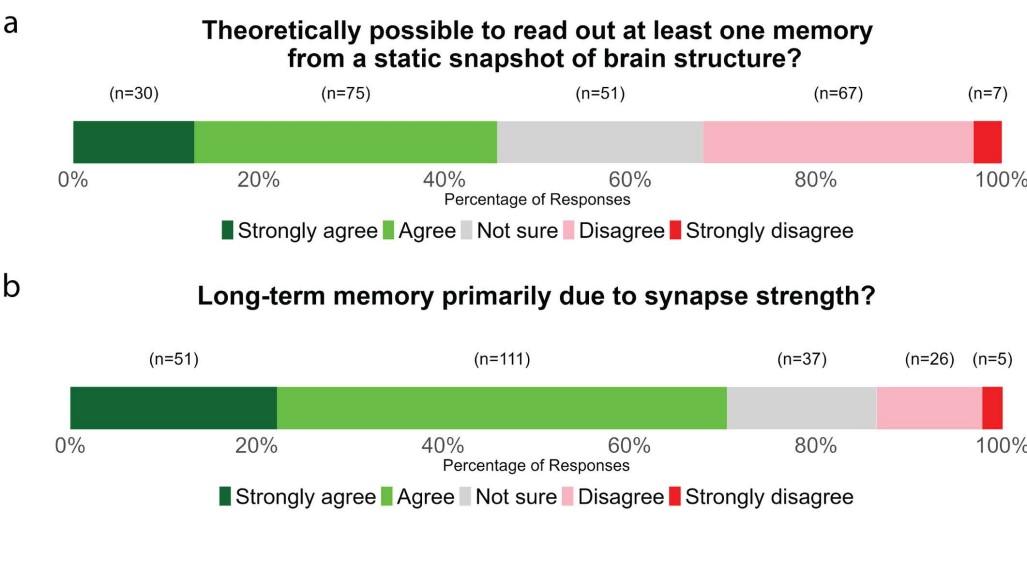

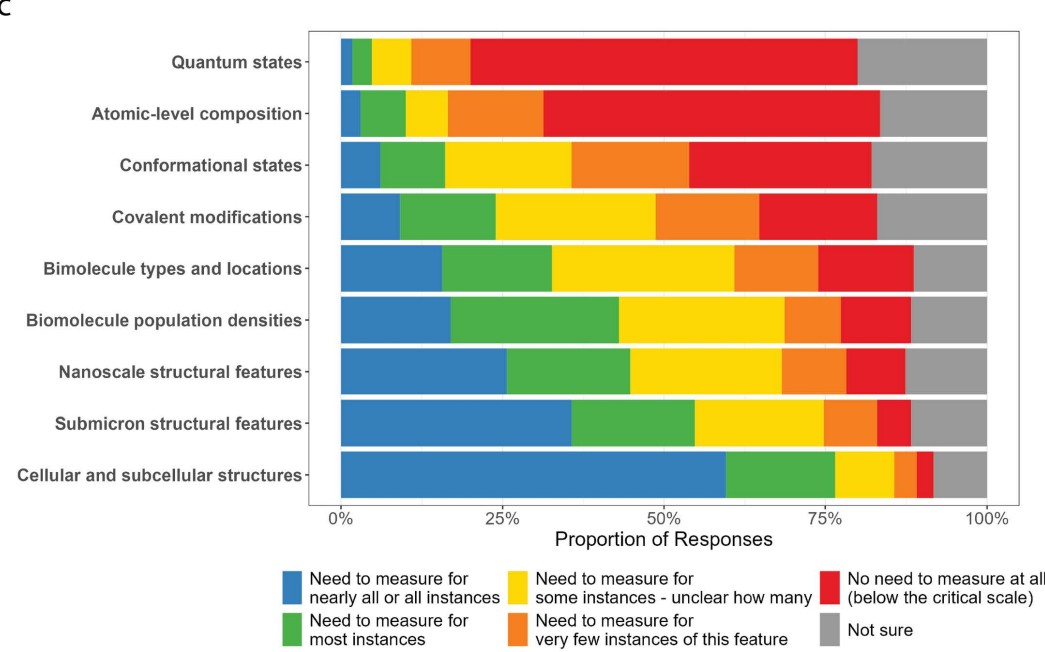

**Fig 1. Beliefs about the physical basis of memory.** a) Responses to the question 'To what extent do you agree with the following statement: "*Given the scientific knowledge assumed in Point A existed, it would be theoretically possible to read out the information corresponding to at least one specific non-trivial memory from a static snapshot of the structure (including biomolecules) of an organism's entire brain.*" b) Responses to the question '*Some neuroscientists have suggested that, while molecular and subcellular details play a role, the majority of information for long-term memories is likely physically stored in the brain at the level of neuronal connectivity patterns and ensembles of synaptic strengths (e.g., Poo et al., 2016). To what extent do you agree with the following statement: "The structural basis of long-term memories primarily consists of lasting changes in neuronal connectivity and ensembles of synaptic strengths, rather than in molecular or subcellular details.*" c) Participants' views on the critical scale for memory encoding in the brain.

and electrical gradients (15.2%) and quantum-level properties (5.2%). A small percentage also selected the option that the paradigm of the question was fundamentally flawed (5.2%).

A prominent review paper on engrams published in 2016 claimed *"There is a clear consensus on where the memory engram is stored - specific assemblies of synapses activated or formed during memory acquisition"* [16]. We sought to determine whether such a consensus truly existed by asking participants whether they agreed that "*The structural basis of long-term memories primarily consists of lasting changes in neuronal connectivity and ensembles of synaptic strengths, rather than in molecular or subcellular details.*" The majority (70.5%) of participants agreed or strongly-agreed (Fig 1b).

We also sought to assess where participants believed the 'critical scale' exists for the physical basis of memory in the brain, by asking them about whether it would be necessary to read out features ranging from the 'quantum states of biomolecules' through to 'cellular and subcellular structures (~ 500 nm resolution)' in order to extract a long-term memory from a static brain snapshot (Fig 1c). The proportion of responses claiming a feature needed to be measured increased monotonically as the physical scale increased, with the threshold where a majority of participants claimed 'at least some instances' needed to be measured occurring at 'Types and locations of individual biomolecules'.

## Implications for memory extraction from brain structure

Differing views about the physical basis of memory have different implications for the practical feasibility of potentially extracting memory-related information from brain structure. Rather than solely considering idealized 'static snapshots' of the brain, we asked participants to provide their subjective probability estimates on whether memory-related information could theoretically be extracted from brains whose structure was preserved using existing techniques, such as aldehyde-stabilized cryopreservation.

Aldehyde-stabilized cryopreservation (ASC) has been shown to preserve the brain's structural connectome, including glial cell morphology and locations [24]. It is also expected to preserve the composition and spatial distribution of nucleic acids, proteins, lipids, and carbohydrates across the brain, capturing a snapshot of the brain's state just prior to the procedure [28]. However, aldehyde-stabilized cryopreservation halts dynamic activity and dissipates electrochemical gradients. Were participants to believe the physical basis of memory was primarily in brain structures such as synaptic connectivity, they would presumably assign a non-trivial probability to the theoretical feasibility of extracting memory-related information from a brain preserved using this technique.

The median subjective probability estimate for the question of whether *"a brain successfully preserved with the aldehyde-stabilized cryopreservation method would retain sufficient information to theoretically decode at least some long-term memories?"* was 41.0% (Fig 2a). Visual inspection of the data suggested that the response distribution was bimodal, with one peak at approximately 75% and another at approximately 10%. This was supported by a Hartigans' dip test for multimodality, after a probit transformation of the data to adjust for boundary conditions of probability estimates at 0 and 100% (D = 0.04, p = 0.011).

It is possible that participants may have provided a high probability to the question of whether a brain preserved "*with the aldehyde-stabilized cryopreservation method would retain sufficient information to theoretically decode at least some long-term memories*" even while thinking that most long-term-memory-related information would be lost in such a scenario. Accordingly, we also asked participants to provide their probability estimates on whether it would ever be theoretically possible to create a 'whole brain emulation' from a brain preserved using aldehyde-stabilized cryopreservation [23]. We defined a whole brain emulation as "*an artificial digital version of a brain …[which] would replicate the internal causal structure of the original brain so closely that its behavioral responses to inputs and outputs would be indistinguishable from the biological version on which it was based.*" As the creation of such an entity would presumably require much more long-term-memory-related information to be available than that merely required to decode at least some long-term memories, assignment of a high probability to this question would indicate strong support for the notion that long-term memories are stored in structural properties of neurophysiology.

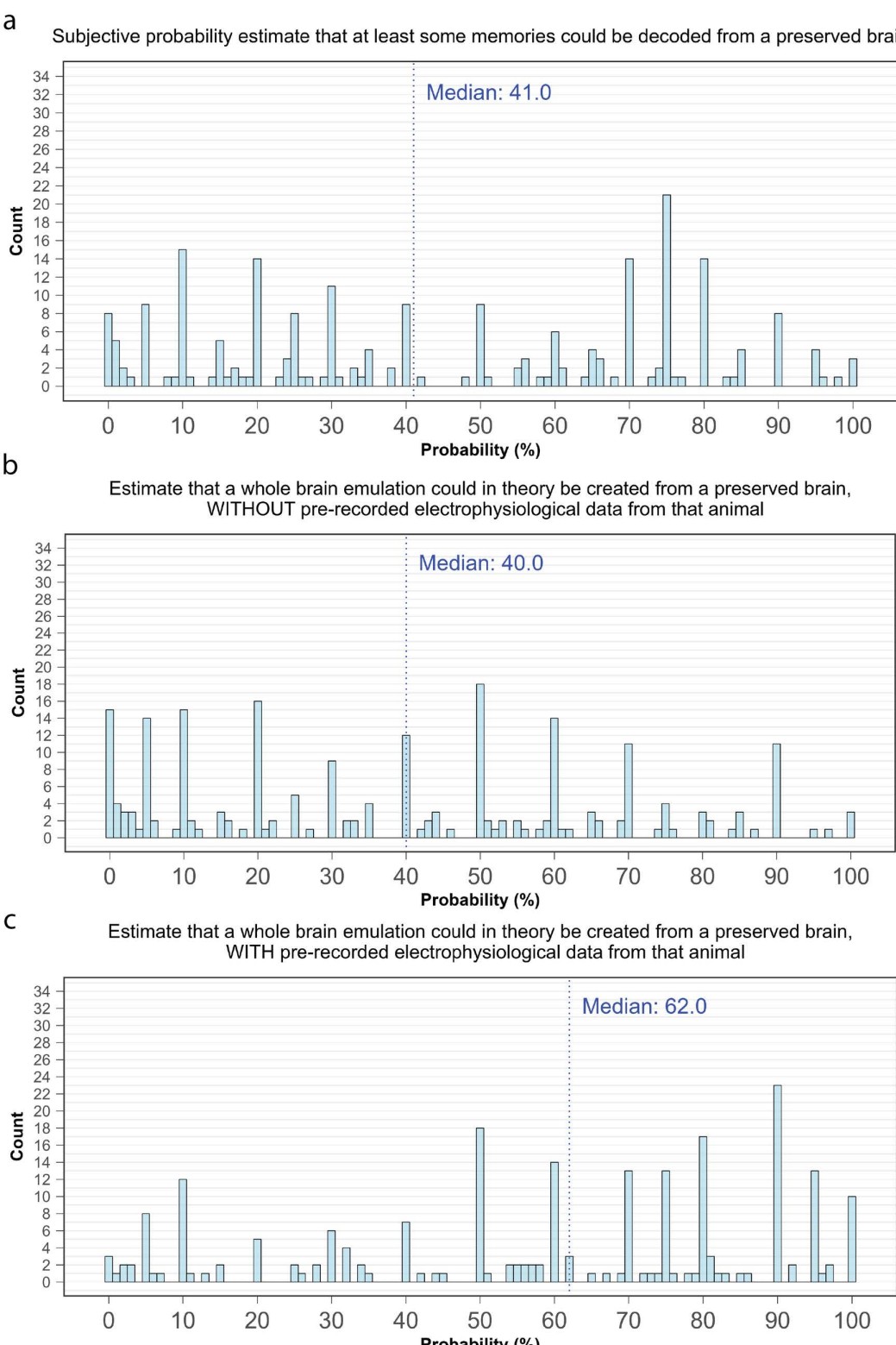

**Fig 2. Subjective probability estimates for the feasibility of memory extraction from a preserved brain.** a) Responses to the question "*Given the scenario outlined above, and assuming that the preserved brain can be maintained indefinitely, what is your subjective probability estimate that a brain*

*successfully preserved with the aldehyde-stabilized cryopreservation method would retain sufficient information to theoretically decode at least some long-term memories?*" b) Responses to the question '*Imagine we had access to a detailed static structural map of an individual animal's brain, such as one obtained through an advanced aldehyde-stabilized cryopreservation procedure. However, instead of having dynamic recordings from that specific individual, we only have access to a large dataset of dynamic brain activity recordings from other members of the same species (including detailed electrophysiological characterisation of all brain cell subtypes). Given this scenario, what is your subjective probability estimate that a whole brain emulation could in theory be created that accurately reproduces the long-term memory behavior of the original brain?*' c) Responses to the question '*Assume we had access to detailed electrophysiological recordings of an individual animal's brain cells, as well as a complete static structural map of that same animal's brain (captured at or below the critical scale). What is your subjective probability estimate that a whole brain emulation could in theory be created that accurately reproduces the long-term memory behavior of the original brain?*'.

The median probability assigned to the question of whether an emulation could theoretically be created from a preserved brain – assuming only generic knowledge of the electrophysiological properties of the neuronal subtypes for the species in question – was 40.0% (Fig 2b). In a different scenario, where active recordings could be taken from the brain in question prior to its preservation, the median probability increased to 62% (Fig 2c). Probabilities assigned in these two scenarios, as well as to the weaker question of memory decoding, were all strongly correlated ('decoding' to 'emulation without pre-recording': $\rho = 0.52$, $p < 10^{-16}$; 'decoding' to 'emulation with pre-recording': $\rho = 0.40$, $p < 10^{-10}$).

## Timeline estimates for the feasibility of whole brain emulation

Even if it is theoretically possible to extract long-term-memory related information by creating an emulation of a preserved brain, there remains the question of how far from contemporary neuroscientific capabilities such a feat would be. To determine what the neuroscientific community's beliefs were on this matter, we asked participants "*by which year there will have existed a whole brain emulation of at least one member of each of the following species (C. elegans, mouse, human)*" and asked them to provide the years for which they were 10%, 50%, and 90% confident this would have been achieved. Participants were asked to round their answer to the closest available year out of '2025', '2035', '2045', '2055', '2065', '2075', '2085', '2095', '2125', '2150', '2200', and 'Never'.

The median participant estimate predicted a *C. elegans* would likely be emulated around 2045 [10%: 2035; 50%: 2045, 90%: 2055], a mouse around 2065 [10%: 2045; 50%: 2065, 90%: 2095], and a human around 2125 [10%: 2075; 50%: 2125, 90%: 2200] (Fig 3).

## Influence of background and expertise

We wanted to know whether experts in researching the neurophysiology of memory had different views on the physical basis of memory from those of the broader neuroscientific community. Focusing on the question of whether "*a brain successfully preserved with the aldehyde-stabilized cryopreservation method would retain sufficient information to theoretically decode at least some long-term memories?*", we found that Engram Experts did have a lower median estimate than that of COSYNE Neuroscientists (30.0% vs 50.0%), although this result was not statistically significant (W = 2007.5, p = 0.10) (Fig 4a).

Rather than splitting the participant data based on cohort allocation, we also sought to ascertain whether participant views differed based on their self-reported primary approach to neuroscience research (i.e., wet-lab, theoretical, both). There were no clear observed effects of research background ($\chi 2(2) = 1.55$, p = 0.46) (Fig 4b). Additionally, there were no significant effects due to education level (S2 and S3 Figs).

However, analysis of correlations between survey responses revealed several significant relationships between respondents' theoretical views and their practical predictions. Participants' estimates for the probability of being able to extract the information required for at least some long-term memories from a brain preserved using aldehyde-stabilized cryopreservation were found to be associated with their theoretical viewpoints (Fig 5). These ASC probability estimates were strongly correlated with respondents' belief in the possibility of whole brain

## Estimated year by which there is an X% probability an emulation for a member of this species will have been successfully created

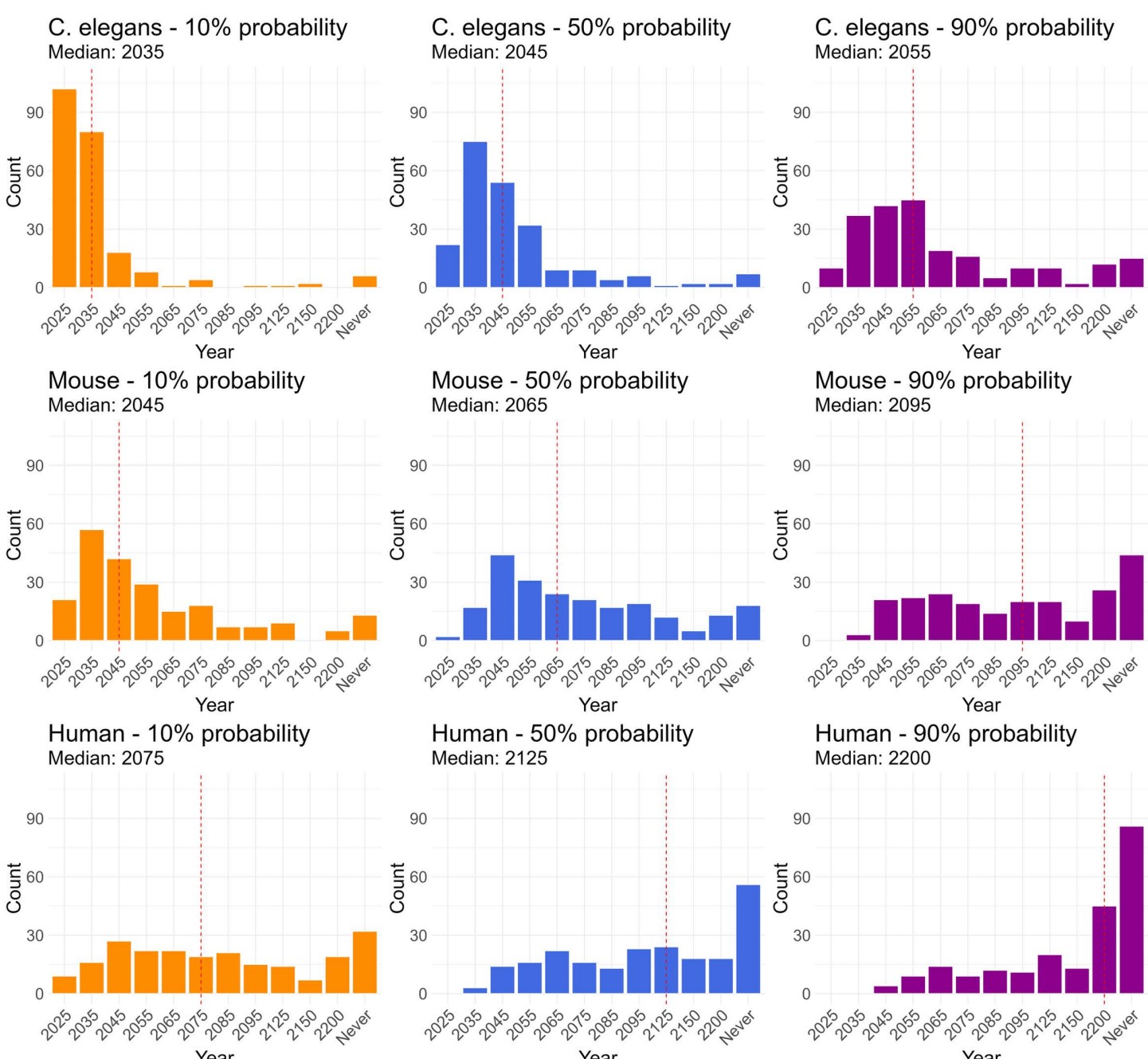

**Fig 3. Estimated years by which there is an X% probability that a whole brain emulation will have been created for at least one member of the *C. elegans*, mouse, and human species.**

a

**Estimate that at least some memories could be decoded from a preserved brain**

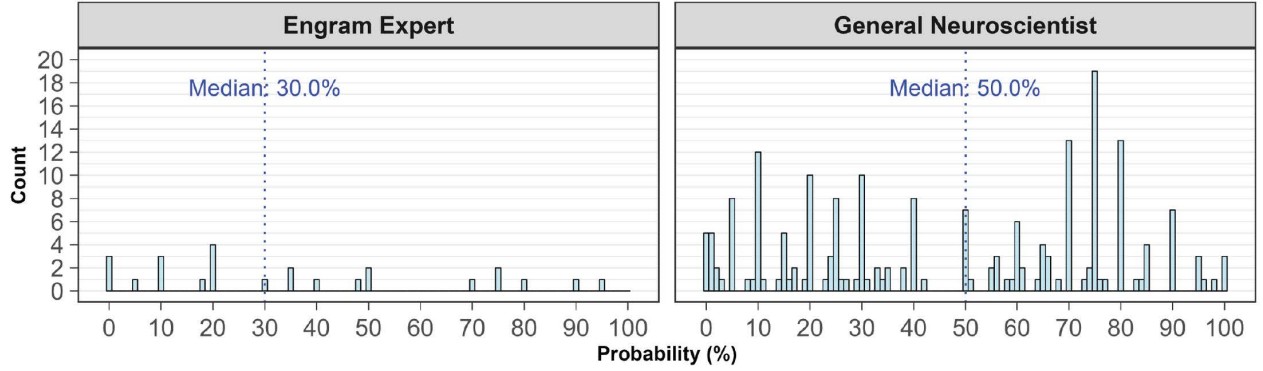

b

**Estimate that at least some memories could be decoded from a preserved brain**

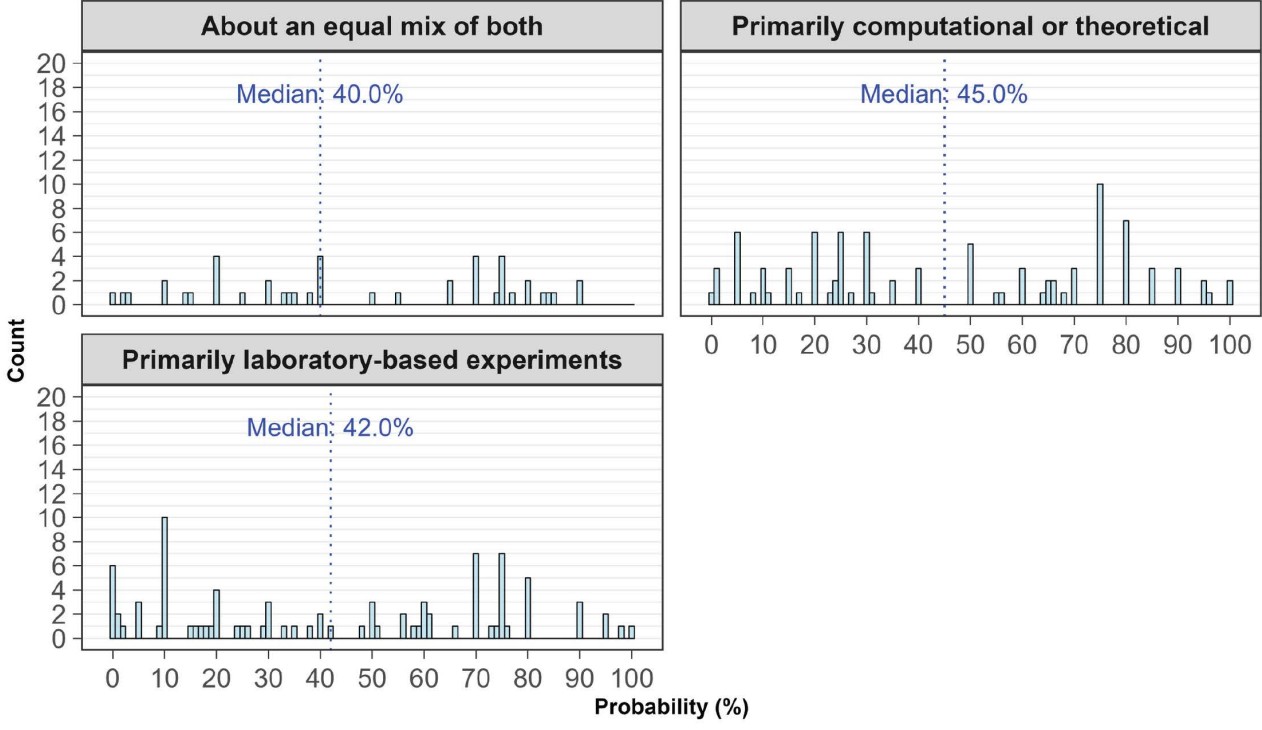

**Fig 4. Subgroup analysis of subjective probability estimates for the feasibility of memory extraction from a preserved brain.** a) Participant data split into either the 'Engram Expert' or 'COSYNE Neuroscientist' cohorts. b) Participant data split based on self-reported response to '*Which of the following best describes your primary approach to neuroscience research?*.

emulation without dynamic recordings ($\rho = 0.52$, $p < 10^{-15}$) and their belief in the theoretical possibility of memory extraction from static brain structure ($\rho = 0.47$, $p < 10^{-12}$). Demographic factors showed modest relationships, with age negatively correlated with ASC probability estimates ($\rho = -0.23$, $p < 10^{-3}$). Notably, neither memory expertise ($\rho = -0.08$, $p = 0.34$) nor preservation expertise ($\rho = -0.01$, $p = 0.96$) nor neural modeling expertise ($\rho = 0.14$,

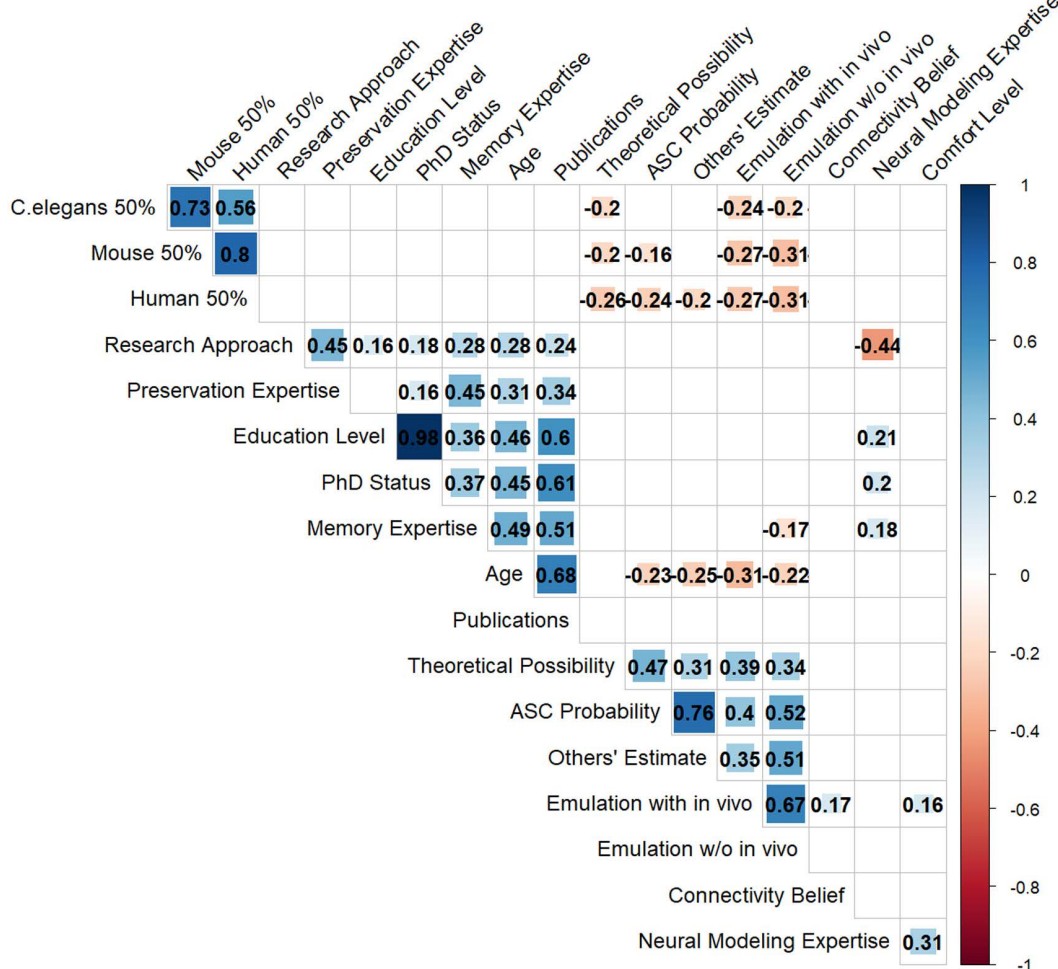

**Fig 5. Correlations between participant responses to different questions.** Only correlations that are statistically significant after Benjamni-Hochberg correction (false discovery rate set to 0.05) are shown. *C.elegans/Mouse/Human 50%* corresponds to the participant's estimated year by which there is a 50% chance an emulation of that animal will have been created. *Research approach* corresponds to their primary approach to research (wet-lab, dry-lab, both). *Preservation expertise* corresponds to their self-reported expertise in 'brain preservation techniques'. *Education level* corresponds to their highest completed degree. *PhD Status* corresponds to whether they had obtained a PhD. *Memory expertise* corresponds to the participant's self-reported expertise in 'neuroscience of memory storage'. *Age* corresponds to the participant's age. *Publications* corresponds to their number of publications. *Theoretical possibility* corresponds to their agreement with whether long-term-memory-related information could be extracted from a static snapshot of the brain. *ASC Probability* corresponds to their subjective probability estimate that long-term memories could be decoded from a brain preserved using ASC. *Other's estimate* corresponds to their estimate of what other neuroscientists will estimate for ASC Probability. *Emulation with in vivo* corresponds to their probability estimate for whether a whole brain emulation could be created from a preserved brain, assuming electrophysiological recordings were taken from the animal prior to preservation. *Emulation w/o in vivo* corresponds to their probability estimate for whether a whole brain emulation could be created from a preserved brain without prior electrophysiological recordings from that animal. *Connectivity Belief* corresponds to their agreement with the statement that 'the structural basis of long-term memories primarily consists of lasting changes in neuronal connectivity…'. *Neural Modeling Expertise* corresponds to their self-reported expertise in 'Neural modeling, simulation, or emulation'. *Comfort level* corresponds to how comfortable they would feel publicly discussing these issues with colleagues.

p = 0.09) showed significant correlations with ASC probability estimates. The strongest relationship was between participants' ASC preservation success estimates and their predictions of other participants' estimates ($\rho = 0.76$, $p < 10^{-42}$), suggesting that participants do not necessarily appreciate the wide view of beliefs that others in the community hold on this topic.

## Discussion

This study provides the first formal survey of the neuroscience community's beliefs on the physical basis of long-term memory. We found that the majority of participants endorse the claim that "*the structural basis of long-term memories primarily consists of lasting changes in neuronal connectivity and ensembles of synaptic strengths, rather than in molecular or subcellular details*". However, there was no clear consensus on exactly which neurophysiological feature or scale is critical for memory storage, with only features at or below the level of 'atomic-level features of biomolecules' being declared 'below the critical scale' by a majority of participants. Correspondingly, there were a broad range of estimates for the probability of whether long-term memories could be decoded from brain structure alone. While the median participant estimate of whether memories could be extracted from a preserved brain was 40%, there was considerable variance and some evidence for a bimodal distribution of results. Similar results were seen for the question of whether a whole-brain emulation could theoretically be created from a preserved brain, with participants estimating even odds it will be possible to create these for a *C. elegans* by 2045, a mouse by 2065, and a human by 2125. No notable effects of participant background or expertise were observed, except for a small negative correlation between self-reported expertise in 'neuroscience of memory storage' and estimates of whether a whole-brain emulation could possibly be created from a preserved brain without prior *in vivo* recordings.

We did not find a unanimous view on which particular neurophysiological features are critical for memory readout. Participants did indicate some boundaries, with a majority stating that the atomic-level composition of biomolecules can be ignored, while subcellular structures at ~500 nm resolution would definitely need to be ascertained. Between these though, no clear finding emerged for which particular features between these two scales are crucial for memory storage.

This is perhaps unsurprising, as despite the progress made by neuroscientists in manipulating memories over the past decade, researchers have noted that the theoretical framework for precisely identifying and understanding engrams is still absent [29]. Demonstrations that long-term memories can survive periods of electrocerebral silence [2], or can be manipulated through optogenetic activation of particular neurons and erasure of particular synapses [12,13,30], provides strong evidence that memory is dependent on structural features of neurophysiology. Nonetheless, when a mouse recalls the route through a maze, it is still unclear if this representation ultimately depends on the specific densities of receptors on particular dendritic spines in particular circuits, the coarse-grained connectivity strength of several populations of neurons, intrinsic plasticity manifested through persistent modifications of voltage-gated ion channels that alter neuronal excitability [8,9], or some combination of these or other neurophysiological features working in parallel. Consider how in genetics, it is understood how DNA triplets (codons) encode protein sequences through the processes of transcription and translation. While memory neuroscientists agree that synapses are crucial, there is no agreement on which features (if any) are equivalent to a codon.

Despite an inability to identify which neurophysiological features are critical for long-term memory storage, a substantial proportion of participants indicated that memories could probably be decoded from the static structure of a preserved brain. This held even to the extent that the median estimate of whether 'a brain preserved by aldehyde-stabilized cryopreservation could potentially be used to create a whole-brain emulation capable of reproducing the long-term memory behavior of an animal' was 40%. This optimistic view on what information is retrievable from structure alone may have been influenced by the recent demonstration that a whole-brain model of *Drosophila*, reconstructed from preserved brain slices imaged using electron microscopy, partially reproduced the animal's natural feeding behavior [31]. While this was merely an observation of instinctual behavior, rather than individual-specific long-term memory, it is possibly suggestive of why only 10.3% of survey respondents indicated there was a less than 10% probability of a human whole-brain emulation ever being achievable. Correspondingly, as the feasibility of creating more sophisticated models of animal brains increases to the point of being able to potentially create a faithful whole brain emulation, it should become easier to test whether an emulation can indeed recreate an animal's long-term memories. If this eventually proves possible, manipulations of the emulation should further clarify which neurophysiological features are actually critical for memory.

Although questions about 'memory extraction from preserved brains' and 'whole brain emulation' are admittedly strange and speculative, we believe they provide a practical framework for understanding memory's physical basis. These concepts create a direct link between theoretical views and their logical consequences. If memories are indeed encoded in stable structural elements like synaptic connectivity patterns - as 70.5% of our respondents indicated - then such memories should theoretically remain accessible in appropriately preserved neural tissue. Alternatively, if memories depend primarily on dynamic processes that cease at preservation, then extraction would be impossible. This approach is analogous to asking electrical engineers about data recovery from powered-down hard drives to understand their beliefs about how data is physically stored. We acknowledge, however, that while these questions provide valuable insights into neuroscientists' views on memory storage mechanisms, they alone do not establish a scientific consensus on the precise physical basis of memory, as evidenced by the diversity of responses we received.

We note that should it become possible to decode memories from a preserved brain, let alone create a whole brain emulation, this would have profound moral, philosophical, and societal consequences. As this survey focused solely on establishing the potential feasibility of this possibility, we asked participants to disregard these considerations when providing their responses. However, the substantial number of neuroscientists who indicated that memories could likely be extracted from preserved brains suggests that examining the ethical implications of this technology should not be dismissed. For example, these include the potential for life extension through means of whole brain emulation [32,33], or privacy violations that could occur through inappropriate memory extraction from a person's preserved brain [34]. Given the median participant response indicated a 10% probability of the first human brain emulation occurring around 2075, and 50% by 2125, these prospects and pitfalls need to be understood in the decades before these capabilities might become available.

There are several limitations with both the data collection and analysis of this study that warrant consideration. First, as we received responses from only a subsample of our target populations (Engram Experts: 12.1%, COSYNE Neuroscientists: 7.3%), our findings may be subject to response bias. However, this possibility is mitigated by the fact that our invitation email only revealed this was a survey on the neuroscience of memory (i.e., it did not provide any detail of the specific questions), as well as the high completion rate (~75%) of those respondents who began the survey.

Second, while we restricted the survey to questions about long term memory, we did not ask participants to provide separate responses for the various subtypes of long term memory. This precludes the possibility of determining whether participants have different views about the physical nature of engrams underlying episodic memory from those that form the substrate of procedural memories.

Third, we note that the COSYNE Neuroscientists comprise only one subsection of the neuroscientific community with a particular focus on computational and systems neuroscience, and that ideally this survey would be replicated with participants drawn from additional neuroscience subfields (such as attendees of the Society for Neuroscience or Federation of European Neuroscience Societies conferences). However, we note that the Engram Experts serve as an internal control, as they were a non-COSYNE cohort of largely behavioural neuroscientists who have published studies on the neuroscience of engrams.

Fourth, in terms of research background, those doing laboratory experiments can be distinguished into multiple types, including wet-lab experiments (patch clamping, animal behaviour, etc.) and neuroimaging techniques, which are substantially different. In future studies, it would be better to break down the participants into finer research background categories.

Fifth, although we attempted to provide background information through the preamble to the questions in our survey, this information was of course incomplete. Respondents have vastly different amounts and sources of knowledge relevant to this topic, which surely affected their responses to the questions.

Sixth, we note that our survey question assumes ASC would provide ideal-quality ultrastructural preservation, which may not be achievable routinely, especially outside laboratory settings, and depends on specific assumptions about ASC

mechanisms. We note that some studies comparing preservation techniques have suggested that there are certain ultra-structural artifacts introduced by aldehyde fixation compared to cryofixed brain tissue [35,36]. These findings are mixed, as more recent work has shown comparable measurements between techniques [37], and the preponderance of evidence suggests that the ultrastructural appearance achieved with both aldehyde fixation and cryofixation is largely similar [38]. A valuable follow-up study would examine whether long-term memories might be preserved under more realistic conditions, such as when people donate their bodies to science. Unlike the ideal conditions assumed in our survey, these situations involve unavoidable brain deterioration due to the cessation of blood flow and associated tissue decomposition. Such circumstances typically include periods of ischemia prior to preservation, potentially resulting in cell death or synaptic loss over time.

Last, we should flag that our analysis of the data is incomplete, as participants provided additional information through hundreds of free-text comments that this report has not examined. Informal exploration of these comments reveals nuanced perspectives that quantitative data alone cannot capture. For instance, participants who agreed that memories are primarily stored in synaptic connectivity sometimes qualified their responses with significant uncertainty, noting it could "go either way" or represented "wild speculation." Others highlighted definitional ambiguities, particularly regarding what constitutes "emulation" or how to interpret terms like "primarily." While we chose not to exclude potentially contradictory responses to avoid introducing selection bias, these comments suggest that many seemingly-contradictory responses reflect sophisticated reasoning rather than misunderstanding. The qualitative insights underscore a central finding of our study: while there may be provisional agreement on broad principles of memory storage, substantial uncertainty remains about specific mechanisms and implications.

In conclusion, this survey reveals that while most neuroscientists believe long-term memories are stored in structural features of the brain, there is no clear consensus on which specific neurophysiological features or scales are critical for memory storage. The finding that a substantial proportion of participants believe memories could theoretically be extracted from preserved brains, despite the aforementioned uncertainty about the precise physical basis of memory, suggests an underlying confidence in the structural nature of memory storage. However, the lack of agreement on critical features and high variance in probability estimates for the feasibility of memory extraction or brain emulation, indicates that fundamental questions about memory's physical basis remain unresolved. Despite this, as technological capabilities advance in the realm of whole brain emulation, we may gain empirical evidence to resolve these theoretical uncertainties. Future studies would benefit from examining different memory subtypes separately, as well as analyzing the rich qualitative data provided by neuroscience experts. Because understanding the physical basis of memory storage has consequences that extend far beyond the laboratory, the ethical and philosophical implications of our observations of the neuroscience community's beliefs should be broadly examined before they become increasingly practically relevant.

## Supporting information

**S1 Fig. Analyzing the demographic attributes of participants who completed the survey vs those who dropped out after the initial demographic section.** There were no differences between these groups based on: a) age (X: 10.384, df = 7, p = 0.17); b) publication count (X: 10.204, df = 5, p = 0.07); c) education level (X: 4.309, df = 3, p = 0.23); d) or research approach (X: 4.518, df = 2, p = 0.10).
(TIF)

**S2 Fig. a) Participants did not differ in their agreement as to whether it would be possible to read out memory from static brain structure based on whether they had obtained a PhD (X = 6.375, df = 5, p = 0.27). b) Participants did not differ in their agreement as to whether long-term memory is primarily due to synapse strength based on whether they had obtained a PhD (X = 2.817, df = 5, p = 0.73).**
(TIF)

**S3 Fig. a) Participant estimates that at least some memories could be decoded from a preserved brain did not differ for those with and without a PhD (W = 5134.5, p = 0.42). b) Participant probability estimates for whether a whole brain emulation could be created from a preserved brain without pre-recorded data did not differ for those with and without a PhD (W = 5234.5, p = 0.94). c) The same was seen if the question was changed to include pre-recorded electrophysiological data (W = 4724, p = 0.22).**
(TIF)

## Acknowledgments

Research funding for this study was supported by a CryoDAO grant. We would like to thank Kenneth Hayworth and John Smart for helpful personal communications.

## Author contributions

**Conceptualization:** Ariel Zeleznikow-Johnston, Emil F. Kendziorra, Andrew T. McKenzie.

**Data curation:** Ariel Zeleznikow-Johnston.

**Formal analysis:** Ariel Zeleznikow-Johnston, Andrew T. McKenzie.

**Funding acquisition:** Ariel Zeleznikow-Johnston, Emil F. Kendziorra.

**Investigation:** Ariel Zeleznikow-Johnston.

**Methodology:** Ariel Zeleznikow-Johnston, Andrew T. McKenzie.

**Project administration:** Ariel Zeleznikow-Johnston.

**Resources:** Ariel Zeleznikow-Johnston.

**Software:** Ariel Zeleznikow-Johnston.

**Validation:** Ariel Zeleznikow-Johnston.

**Visualization:** Ariel Zeleznikow-Johnston.

**Writing – original draft:** Ariel Zeleznikow-Johnston.

**Writing – review & editing:** Ariel Zeleznikow-Johnston, Emil F. Kendziorra, Andrew T. McKenzie.

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
