## [Decision Letter · Decision Letter 0]

PONE-D-25-09935What are memories made of? A survey of neuroscientists on the structural basis of long-term memoryPLOS ONE

Dear Dr. Zeleznikow-Johnston,

Thank you for submitting your manuscript to PLOS ONE. After careful consideration, we feel that it has merit but does not fully meet PLOS ONE’s publication criteria as it currently stands. Therefore, we invite you to submit a revised version of the manuscript that addresses all the points raised during the review process. In particular, the need to discuss intrinsic plasticity, as a potential cellular mechanism for memory formation is an important point.

We look forward to receiving your revised manuscript.

Kind regards,

Stéphane Charpier

Academic Editor

PLOS ONE

Journal Requirements:

3. Thank you for stating the following financial disclosure: [Research funding for this study was supported by a CryoDAO grant (2024.1)]. 

4. Thank you for stating the following in the Competing Interests section: [I have read the journal's policy and the authors of this manuscript have the following competing interests: Andrew McKenzie is an employee of Oregon Brain Preservation, a non-profit brain preservation organization, and a director of Apex Neuroscience, a non-profit research organization. Emil Kendziorra is a shareholder and CEO of Tomorrow Biostasis, a biostasis provider, President of the Board of the European Biostasis Foundation, a non-profit research foundation, a shareholder and director at Oxford Cryotechnology, Inc., a cryopreservation research organization, and a board member at CryoDAO, a Swiss research association.].

5. Please update your submission to use the PLOS LaTeX template. The template and more information on our requirements for LaTeX submissions can be found at http://journals.plos.org/plosone/s/latex .

Reviewers' comments:

Reviewer's Responses to Questions

**Comments to the Author**

1. Is the manuscript technically sound, and do the data support the conclusions?

Reviewer #1: Yes

Reviewer #2: Partly

2. Has the statistical analysis been performed appropriately and rigorously? 

Reviewer #1: Yes

Reviewer #2: I Don't Know

3. Have the authors made all data underlying the findings in their manuscript fully available?

Reviewer #1: Yes

Reviewer #2: Yes

4. Is the manuscript presented in an intelligible fashion and written in standard English?

Reviewer #1: Yes

Reviewer #2: Yes

5. Review Comments to the Author

Reviewer #1: This manuscript aims to gather neuroscientists' opinions on the structural basis of memory storage and to provide a detailed analysis of the collected data. This study indicates that the main physical support of memory is synaptic plasticity.

The paper is very clear, well written and informative. I have only relatively minor points.

Minor point:

1. Figure 1C, 5th raw: please correct Bimolecule by Biomolecule.

2. Page 18, 3rd line: changes and not chances

3. It is not clear in the manuscript what other mechanism could support the formation of memory. Are the authors thinking at intrinsic plasticity that is due to changes in voltage-gated ion channels and that is generally associated with synaptic changes (see for instance Titley et al., 2017 and Debanne et al., 2019)? A brief discussion of these aspects appears necessary.

Reviewer #2: The article of Zeleznikow-Johnston, Kendziorra and McKenzie present the results of a survey they conducted on a sample of neuroscientists. The main goal of the study is to evaluate whether neuroscientists believe human brain can be stored, after fixation, in its native state ; and whether neuroscientists believe it is possible to access the content of a memory by looking at the fixed brain neuronal network.

The authors claims the goal of the survey is to establish whether the idea that memory is stored in patterns of patterns is a scientific consensus.

But the survey strangely goes into questions about emulation of a whole brain, reading memory content from a fixed brain.

One understands the reason behind a survey with such surprising questions when considering the conflict of interest statement. The authors are engaged in companies that offer the following service : store your brain after death, with the promise to read or do something from that preserved brain.

This reviewer is therefore in between two minds. On the one hand, the text of the article respects the rigor of science, and provides objective analysis, on the other hand the questions used in the survey do not assess the presented goal of evaluating a scientific consensus.

More precisely :

There is a problem with the sampling. The authors name « general neuroscientists » people who attended COSYNE congress in years 2022/23/24. Looking at speakers and titles of posters from the congress those three years, it appears the term « general neuroscientists » does not apply. It would apply when considering people who attended SFN or FENS.

COSYNE is clearly oriented towards network, computation, theoretical neuroscience. With some exception genetic and molecular levels are not represented, and experimental recordings are implemented at the level of the network, rather that single cell. There is also overrepresentation (as compared to what would be general neuroscientists) of computational and model neuroscience, which is a strong bias when question such as « could we emulate a brain in a computer » or  « could we read information from the knowledge of the structure of a network » is given.

At minimum, the term « general neuroscientists » should be changed. Information on the orientation of COSYNE congress should appear in the article.

Asking “Given the scientific knowledge assumed in Point A existed, it would be theoretically possible to read out the information corresponding to at least one specific non-trivial memory from a static snapshot of the structure (including biomolecules) of an organism's entire brain.” to scientists attending SFN would give very different result.

On that question, one may also possibly expect that some people drop out of the survey when reading that question because they consider it is ill-defined (note that the question appears as the first question of the survey -not mentioning the 4 preeceding questions on the profile of the scientists such as age and level of education). This may represent a strong bias ?

Same for the question “The structural basis of long-term memories primarily consists of lasting changes in neuronal connectivity and ensembles of synaptic strengths, rather than in molecular or subcellular details.”, people at SFN may not be 70 % to agree.

The categorization of the sample into either « Primarily computational or theoretical (e.g. dry-lab experiments » or « Primarily laboratory-based experiments (e.g., wet-lab experiments, neuroimaging, etc. » is not satisfactory. Neuroimaging is so far from wet-lab experiments, people doing neuroimaging are often from cognitive science.

102 out of 312 « scientists » were at Bachelor or Master degree. Mixing students and researchers seems strange choice. How that population responded in comparison to the one having a PhD ? The time of PhD, and research time after obtaining the PhD, is important for one realizes better what is feasible and what science and technology can indeed bring. Please provide an analysis. And I do not see the level of eductaion in the correlation chart of Fig5.

There are wrong and biased statements in the article.

1/ I am puzzled with the argument

« Long-term memories can be recalled even after periods of prolonged global neuronal depolarization and inactivity, as can occur in deep hypothermic circulatory arrest (2,3), or despite temporary protein synthesis inhibition (4). »

First, the ref 4, in its abstract, states : « Evidence from learning curves, examination of short-term retention, and posttraining drug injection indicate that initial acquisition is not dependent on such synthesis, but it appears that protein synthesis, during or shortly after training, is an essential step in the formation of long-term memory. » So it states the opposite for what the authors cites it.

Second, why assuming from hypothermia it is evidence the electrophysiological and molecular level is not required ? If cooling, enzymatic activity will stop…and start again after temperature is back to physiological level. Electrophysiological activity as well.

If you wish to maintain your argument, please elaborate, and include and discuss the fact that cooling make disappear dendritic spines, whihc reappear when warming after cooling

Kirov SA, Petrak LJ, Fiala JC, Harris KM. Dendritic spines disappear with chilling but proliferate excessively upon rewarming of mature hippocampus. Neuroscience. 2004;127(1):69-80. doi: 10.1016/j.neuroscience.2004.04.053. PMID: 15219670.

2/ The question “a brain successfully preserved with the aldehyde-stabilized cryopreservation method would retain sufficient information to theoretically decode at least some long-term memories?” may suffer from a bias

In the survey, the introduction before the question gives « If the procedure is successful, the combination of chemical fixation and cryopreservation is expected to allow stable, long-term storage of brains with minimal ultrastructural distortion. »

It seems to me the survey make a statement about the success of the method, and then ask the participant whether he or she believes in its success.

What if the introduction would have added the fact that learning and memory implies synaptic plasticity, which embbed morphological changes of synapses, and the fact that two fixation methods, aldehyde perfusion and cryo fixation, reveals different morphologies for the synapses ?

Korogod N, Petersen CC, Knott GW. Ultrastructural analysis of adult mouse neocortex comparing aldehyde perfusion with cryo fixation. Elife. 2015 Aug 11;4:e05793. doi: 10.7554/eLife.05793. PMID: 26259873; PMCID: PMC4530226.

Tamada H, Blanc J, Korogod N, Petersen CC, Knott GW. Ultrastructural comparison of dendritic spine morphology preserved with cryo and chemical fixation. Elife. 2020 Dec 4;9:e56384. doi: 10.7554/eLife.56384. PMID: 33274717; PMCID: PMC7748412.

What if the introduction would remind to the participant the necessary time delay between postmortem brain extraction (the human subject will not stay next to the perfusion system, waiting for her or his death) and the perfusion ? It implies ischemia and cell death as well as synaptic loss.

In the survey, one could ask if, assuming preservation is ideal, memory content could be read form fiuxed brain ; but the question was « Given the scenario outlined above, and assuming that the preserved brain can be maintained indefinitely, what is your subjective probability estimate that a brain successfully preserved with the aldehyde-stabilized cryopreservation method would retain sufficient information to theoretically decode at least some long-term memories? »

Either participant would answer based on their knowledge, or based on the « scenario », but the scenario offered incomplet information.

I did not understood the correlation analysis. How to correlate « research approach » with « memory expertise » ? There are 2 research approaches (wet and dry experiment oriented) and 2 expertise (engramm expert and general neuroscientist), so 4 correlations can be calculated, rather than 1 ?

I found that to ask to each participant what he or she expect the other would answer is very interesting approach.

Reading the raw data in the .csv table, some problems appear. It is minor but I wonder how the authors considered it for the analysis.

To what extent do you agree with the following statement: “Given the scientific knowledge assumed in Point A existed, it would be theoretically possible to read out the information corresponding to at least one specific non-trivial memory from a static snapshot of the structure (including biomolecules) of an organism's entire brain.”

is followed by

If you think that the above statement would most likely not be possible, even in theory, what additional information beyond a static snapshot do you think would be required to read out a specific memory? (Select all that apply)_Not applicable, I do think this would most likely be possible in theory

a participant (line 6) selected : Strongly disagree

and then : Not applicable, I do think this would most likely be possible in theory

a participant (line 67) selected : Disagree

and then Not applicable, I do think this would most likely be possible in theory

a participant (line 205) selected : Agree

and add a general comment :

The question is ill-posed. In theory, with the right initial conditions, the locations and states of all the matter in the brain, and its coupling with the body is "all there is". But it isn't practical to reconstruct any of this. All known preservation and measurement processes would likely destroy the state irreparably.

Hence these 3 participant gave paradoxical answers

On wether the participants agree Poo’s sentence ("The structural basis of long term memories primarily consists of lasting changes in neuronal connectivity and ensembles of synaptic strengths, rather than in molecular or subcellular details." )

line 130 stated

Agree

and add the comment :Again, wild speculation for now.

Line 169 stated

Agree

and add the comment : I could totally see it going either way, though. It may be that as a computational modeler, I *want* memory to be stored in connectivity, rather than molecular detail.

« primarily » is a problem, two examples illustrate :

line 250 stated : Agree

and add the comment : I agree but I doubt that ongoing neural activity can be completely marginalized out such that only the connectivity matters

line 266 stated : Agree

and add the comment : I think this is quite likely given the current state of neuroscience knowledge, and it is the underlying tacit assumption of much current work in neuroscience. However, the brain frequently proves to be more complex than we thought, so it's difficult to exclude the possibility that more molecular or cell-autonomous mechanisms also play an important role.

When asked to estimate by which year a whole brain will be emulated, 4 participant sgave rather near years (such as 2025, 2035), while adding as comment :

272 not sure what degree of recreation you intend to achieve.

254 I am not sure what "emulation" means. Aren't there already efforts to do this and they consistently suffer from not having any reasonable measure of success.

143 this is a complicated question to answer: what is a "whole brain emulation"? Arguably this already exists (badly) for C. elegans, at least in terms of the electrophysiology.

97 Again it is very unclear to me how you would quantify a success in such a model of "emulation".

How such paradox in answers were treated in the analysis ?

6. PLOS authors have the option to publish the peer review history of their article (what does this mean? ). If published, this will include your full peer review and any attached files.

**Do you want your identity to be public for this peer review?** For information about this choice, including consent withdrawal, please see our Privacy Policy .

Reviewer #1: No

Reviewer #2: No

---

## [Author Response · Author response to Decision Letter 1]

15 May 2025

NB: Please see the attached 'PLOS ONE Response to Reviewers - What are memories made of.pdf' for a nicely formatted version of this response

We thank the reviewers for their helpful comments. Below, we provide detailed responses.

Reviewer #1: This manuscript aims to gather neuroscientists' opinions on the structural basis of memory storage and to provide a detailed analysis of the collected data. This study indicates that the main physical support of memory is synaptic plasticity.

The paper is very clear, well written and informative. I have only relatively minor points.

We thank the reviewer for their many kind comments!

Minor point:

1. Figure 1C, 5th raw: please correct Bimolecule by Biomolecule.

Corrected.

2. Page 18, 3rd line: changes and not chances

Corrected.

3. It is not clear in the manuscript what other mechanism could support the formation of memory. Are the authors thinking at intrinsic plasticity that is due to changes in voltage-gated ion channels and that is generally associated with synaptic changes (see for instance Titley et al., 2017 and Debanne et al., 2019)? A brief discussion of these aspects appears necessary.

We thank the reviewer for this comment. We agree about the need to be more explicit about what we mean by structural vs non-structural mechanisms, as well as to include intrinsic plasticity in the list of discussed memory-relevant neurophysiological mechanisms.

In paragraph 3 of the Introduction section, we have clarified structural vs non-structural properties by adding sentences stating:

“By 'structural,' we refer to relatively stable physical changes to neural components that persist independently of ongoing neural activity or metabolic processes, though these changes may initially require activity-dependent processes to be established. These structural changes exist on a spectrum from modifications of individual proteins and channel properties through to larger-scale alterations in neural connectivity.”

We mean to differentiate these structural properties from things like persistent electrical activity or transient, short-lived, non-structural protein synthesis, which are known to sustain working or short-term memory.

Additionally, we have also added intrinsic plasticity into the list of potential structural mechanisms listed in paragraph 4. As the mechanisms mediating intrinsic plasticity involve features such as voltage-gated channel insertion into the membrane, post-translational modification of these receptors, and alternate splicing, we have deemed these ‘structural’ (in opposition to the transient factors mentioned above).

Lastly, we have also added sentences to the third paragraph of the Discussion section stating that:

“ Nonetheless, when a mouse recalls the route through a maze, it is still unclear if this representation ultimately depends on the specific densities of receptors on particular dendritic spines in particular circuits, the coarse-grained connectivity strength of several populations of neurons, intrinsic plasticity manifested through persistent modifications of voltage-gated ion channels that alter neuronal excitability (8,9), or some combination of these or other neurophysiological features working in parallel.”

Reviewer #2: The article of Zeleznikow-Johnston, Kendziorra and McKenzie present the results of a survey they conducted on a sample of neuroscientists. The main goal of the study is to evaluate whether neuroscientists believe human brain can be stored, after fixation, in its native state ; and whether neuroscientists believe it is possible to access the content of a memory by looking at the fixed brain neuronal network.

The authors claims the goal of the survey is to establish whether the idea that memory is stored in patterns of patterns is a scientific consensus.

But the survey strangely goes into questions about emulation of a whole brain, reading memory content from a fixed brain.

One understands the reason behind a survey with such surprising questions when considering the conflict of interest statement. The authors are engaged in companies that offer the following service : store your brain after death, with the promise to read or do something from that preserved brain.

This reviewer is therefore in between two minds. On the one hand, the text of the article respects the rigor of science, and provides objective analysis, on the other hand the questions used in the survey do not assess the presented goal of evaluating a scientific consensus.

We thank the reviewer for this comment. In order to explain why we have asked these questions, which we agree might be confusing to a reader, we have added the following paragraph to the Discussion section:

“Although questions about ‘memory extraction from preserved brains’ and ‘whole brain emulation’ are admittedly strange and speculative, we believe they provide a practical framework for understanding memory's physical basis. These concepts create a direct link between theoretical views and their logical consequences. If memories are indeed encoded in stable structural elements like synaptic connectivity patterns - as 70.5% of our respondents indicated - then such memories should theoretically remain accessible in appropriately preserved neural tissue. Alternatively, if memories depend primarily on dynamic processes that cease at preservation, then extraction would be impossible. This approach is analogous to asking electrical engineers about data recovery from powered-down hard drives to understand their beliefs about how data is physically stored. We acknowledge, however, that while these questions provide valuable insights into neuroscientists' views on memory storage mechanisms, they alone do not establish a scientific consensus on the precise physical basis of memory, as evidenced by the diversity of responses we received.”

More precisely :

There is a problem with the sampling. The authors name « general neuroscientists » people who attended COSYNE congress in years 2022/23/24. Looking at speakers and titles of posters from the congress those three years, it appears the term « general neuroscientists » does not apply. It would apply when considering people who attended SFN or FENS.

COSYNE is clearly oriented towards network, computation, theoretical neuroscience. With some exception genetic and molecular levels are not represented, and experimental recordings are implemented at the level of the network, rather that single cell. There is also overrepresentation (as compared to what would be general neuroscientists) of computational and model neuroscience, which is a strong bias when question such as « could we emulate a brain in a computer » or « could we read information from the knowledge of the structure of a network » is given.

At minimum, the term « general neuroscientists » should be changed. Information on the orientation of COSYNE congress should appear in the article.

We thank the reviewer for this comment. We agree that the ideal version of this survey would involve surveying a substantial fraction of the participants at SFN or FENS, representing many thousands of neuroscientists from diverse subfields. While this is something we would like to perform in the future, unfortunately at present there is no publicly accessible database for contacting these participants. In contrast, COSYNE conference attendees are publicly listed in the conference abstract booklets, along with their email addresses, which made it possible to contact this entire cohort of participants in a systematic fashion.

Based on your comment, we have changed “general neuroscientists” to “COSYNE neuroscientists” throughout the manuscript, so as to be more precise about where this participant group originates from. We have also added an additional sentence to the methods where the participant cohorts are described, clarifying that ‘COSYNE attendees are self-described as those interested in “the exchange of empirical and theoretical approaches to problems in systems neuroscience”’. We have also added a description in the limitations section of the discussion stating that “Third, we note that the COSYNE Neuroscientists comprise only one subsection of the neuroscientific community with a particular focus on computational and systems neuroscience, and that ideally this survey would be replicated with participants drawn from additional neuroscience subfields (such as attendees of the Society for Neuroscience or Federation of European Neuroscience Societies conferences). However, we note that the Engram Experts can serve as an internal control, as they were a non-COSYNE cohort of largely behavioural neuroscientists who have published studies on the neuroscience of engrams.”

Asking “Given the scientific knowledge assumed in Point A existed, it would be theoretically possible to read out the information corresponding to at least one specific non-trivial memory from a static snapshot of the structure (including biomolecules) of an organism's entire brain.” to scientists attending SFN would give very different result.

On that question, one may also possibly expect that some people drop out of the survey when reading that question because they consider it is ill-defined (note that the question appears as the first question of the survey -not mentioning the 4 preeceding questions on the profile of the scientists such as age and level of education). This may represent a strong bias ?

We thank the reviewer for this comment. To partially address these concerns around potential bias, we assessed whether participants who dropped out after completing the initial demographics section differed in their demographic profile from those participants who completed the entire survey. We have added this additional analysis as a Supplementary Analysis to the manuscript, and added a description of this in the Methods section: “Participants who completed only the demographic section of the survey did not have a statistically significant difference in their demographic profile from those who completed the entire survey (Sup. Fig. 1). Specifically, completers vs non-completers did not differ significantly in terms of age (X: 10.384, df = 7, p = 0.17); b) publication count (X: 10.204, df = 5, p = 0.07); c) education level (X: 4.309, df = 3, p = 0.23); d) or research approach (X: 4.518, df = 2, p=0.10).”

Same for the question “The structural basis of long-term memories primarily consists of lasting changes in neuronal connectivity and ensembles of synaptic strengths, rather than in molecular or subcellular details.”, people at SFN may not be 70 % to agree.

The categorization of the sample into either « Primarily computational or theoretical (e.g. dry-lab experiments » or « Primarily laboratory-based experiments (e.g., wet-lab experiments, neuroimaging, etc. » is not satisfactory. Neuroimaging is so far from wet-lab experiments, people doing neuroimaging are often from cognitive science.

We thank the reviewer for this comment. We have added this as a limitation of the study in the Discussion section: “Fourth, in terms of research background, those doing laboratory experiments can be distinguished into multiple types, including wet-lab experiments (patch clamping, animal behaviour, etc.) and neuroimaging techniques, which are substantially different. In future studies, it would be better to break down the participants into finer research background categories.”

102 out of 312 « scientists » were at Bachelor or Master degree. Mixing students and researchers seems strange choice. How that population responded in comparison to the one having a PhD ? The time of PhD, and research time after obtaining the PhD, is important for one realizes better what is feasible and what science and technology can indeed bring. Please provide an analysis. And I do not see the level of eductaion in the correlation chart of Fig5.

We thank the reviewer for this comment. We have added this requested analysis to the Supplementary Analysis section, and noted this in the ‘Influence of background and expertise’ part of the Results section.

Supplementary Figure 2 displays how participants did not differ in their agreement as to whether it would be possible to read out memory from static brain structure based on whether they had obtained a PhD (X = 6.375, df = 5, p = 0.27), nor in their agreement as to whether long-term memory is primarily due to synapse strength based on whether they had obtained a PhD (X=2.817, df = 5, p = 0.73).

Supplementary Figure 3 displays how participant estimates that at least some memories could be decoded from a preserved brain did not differ for those with and without a PhD (W = 5134.5, p = 0.42). Additionally, participant probability estimates for whether a whole brain emulation could be created from a preserved brain without pre-recorded data did not differ for those with and without a PhD (W = 5234.5, p = 0.94), and the same was seen if the question was changed to include pre-recorded electrophysiological data (W = 4724, p=0.22)

Thank you for pointing out our error in having accidentally left education level out of the correlation matrix chart in Figure 5 - we have corrected this.

There are wrong and biased statements in the article.

1/ I am puzzled with the argument

« Long-term memories can be recalled even after periods of prolonged global neuronal depolarization and inactivity, as can occur in deep hypothermic circulatory arrest (2,3), or despite temporary protein synthesis inhibition (4). »

First, the ref 4, in its abstract, states : « Evidence from learning curves, examination of short-term retention, and posttraining drug injection indicate that initial acquisition is not dependent on such synthesis, but it appears that protein synthesis, during or shortly after training, is an essential step in the formation of long-term memory. » So it states the opposite for what the authors cites it.

We thank the reviewer for this comment. We agree that this sentence and citation is confusing, because we did not clearly define at the outset what we meant by the difference between ‘memory formation’ and ‘memory recall.’ To clarify this point, we have added text to the Introduction section just after this sentence to explain more clearly the point we were trying to make: “Additionally, while a temporary period of protein synthesis inhibition disrupts memory formation when administered during or shortly after training, it has been found to have no effect on memory recall when administered six or more days after training (4). This temporal distinction in the mechanisms between memory formation and recall suggests that long-term memory storage depends on stable structural changes rather than ongoing protein synthesis or dynamic neural activity, unlike working and short-term memory.”

Second, why assuming from hypothermia it is evidence the electrophysiological and molecular level is not required ? If cooling, enzymatic activity will stop…and start again after temperature is back to physiological level. Electrophysiological activity as well.

If you wish to maintain your argument, please elaborate, and include and discuss the fact that cooling make disappear dendritic spines, whihc reappear when warming after cooling

Kirov SA, Petrak LJ, Fiala JC, Harris KM. Dendritic spines disappear with chilling but proliferate excessively upon rewarming of mature hippocampus. Neuroscience. 2004;127(1):69-80. doi: 10.1016/j.neuroscience.2004.04.053. PMID: 15219670.

We thank the reviewer for this comment. We clarified our point about deep hypothermic circulatory arrest in the Introduction section: “Long-term memories can be recalled even after periods of prolonged global neuronal depolarization and inactivity, for example as can occur in deep hypothermic circulatory arrest (2,3). This suggests that ongoing global electrophysiological activity is not required for the retention of already formed long-term memories.”

We also appreciate your reference to Kirov et al. (2004), which we agree is highly relevant to our manuscript. We believe that this reference is not only relevant to hypothermia, but also a good example of our broader point about structural resilience: despite temporary disruption

---

## [Decision Letter · Decision Letter 1]

What are memories made of? A survey of neuroscientists on the structural basis of long-term memory

PONE-D-25-09935R1

Dear Dr. Zeleznikow-Johnston,

We’re pleased to inform you that your manuscript has been judged scientifically suitable for publication and will be formally accepted for publication once it meets all outstanding technical requirements.

Kind regards,

Stéphane Charpier

Academic Editor

PLOS ONE

Additional Editor Comments (optional):

Reviewers' comments:

Reviewer's Responses to Questions

**Comments to the Author**

1. If the authors have adequately addressed your comments raised in a previous round of review and you feel that this manuscript is now acceptable for publication, you may indicate that here to bypass the “Comments to the Author” section, enter your conflict of interest statement in the “Confidential to Editor” section, and submit your "Accept" recommendation.

Reviewer #1: All comments have been addressed

Reviewer #2: All comments have been addressed

2. Is the manuscript technically sound, and do the data support the conclusions?

Reviewer #1: Yes

Reviewer #2: Yes

3. Has the statistical analysis been performed appropriately and rigorously? 

Reviewer #1: Yes

Reviewer #2: I Don't Know

4. Have the authors made all data underlying the findings in their manuscript fully available?

Reviewer #1: Yes

Reviewer #2: Yes

5. Is the manuscript presented in an intelligible fashion and written in standard English?

Reviewer #1: Yes

Reviewer #2: Yes

6. Review Comments to the Author

Reviewer #1: The points raised in the previous round of evaluation have all been satisfactorily addressed. I have no further comment.

Reviewer #2: The authors answered to my comments, and I appreciate that the text of the manuscript was clarified. Some concerns remain, but are subject of debate for the community, rather than issues within the manuscript. The level of transparency with the revised version will allow the debate to take place, for whoever is interested, given the clarification of issues from the first version.

7. PLOS authors have the option to publish the peer review history of their article (what does this mean? ). If published, this will include your full peer review and any attached files.

**Do you want your identity to be public for this peer review?** For information about this choice, including consent withdrawal, please see our Privacy Policy .

Reviewer #1: No

Reviewer #2: No

---

## [Editor Report · Acceptance letter]

PONE-D-25-09935R1

PLOS ONE

Dear Dr. Zeleznikow-Johnston,

I'm pleased to inform you that your manuscript has been deemed suitable for publication in PLOS ONE. Congratulations! Your manuscript is now being handed over to our production team.

Kind regards,

on behalf of

Pr. Stéphane Charpier

Academic Editor

PLOS ONE